# Estimating Wildlife Density as a Function of Environmental Heterogeneity Using Unmarked Data

**Thomas Connor** [1,*], **Wildlife Division** [2], **Emilio Tripp** [2], **William T. Bean** [3], **B. J. Saxon** [2], **Jessica Camarena** [2], **Asa Donahue** [2], **Daniel Sarna-Wojcicki** [1], **Luke Macaulay** [1], **William Tripp** [2] **and Justin Brashares** [1]

[1] Department of Environmental Science, Policy, and Management, University of California, Berkeley, CA 94720, USA; dsarna@berkeley.edu (D.S.-W.); luke.macaulay@berkeley.edu (L.M.); brashares@berkeley.edu (J.B.)

[2] Karuk Tribe, Department of Natural Resources, P.O. Box 282, Orleans, CA 95556, USA; wildlifeteam@karuk.us (W.D.); etripp@karuk.us (E.T.); bsaxon@karuk.us (B.J.S.); jcamarena@karuk.us (J.C.); adonahue@karuk.us (A.D.); btripp@karuk.us (W.T.)

[3] Biological Sciences Department, College of Science and Mathematics, San Luis Obispo, CA 93407, USA; wtbean@calpoly.edu

[*] Correspondence: connort@berkeley.edu

**Abstract:** Recent developments to spatial-capture recapture models have allowed their use on species whose members are not uniquely identifiable from photographs by including individual identity as a latent, unobserved variable in the model. These 'unmarked' spatial capture recapture (uSCR) models have also been extended to presence-absence data and modified to allow categorical environmental covariates on density, but a uSCR model, which allows fitting continuous environmental covariates to density, has yet to be formulated. In this paper, we fill this gap and present an extension to the uSCR modeling framework by modeling animal density on a discrete state space as a function of continuous environmental covariates and investigate a form of Bayesian variable selection to improve inference. We used an elk population in their winter range within Karuk Indigenous Territory in Northern California as a case study and found a positive credible effect of increasing forb/grass cover on elk density and a negative credible effect of increasing tree cover on elk density. We posit that our extensions to uSCR modeling increase its utility in a wide range of ecological and management applications in which spatial counts of wildlife can be derived and environmental heterogeneity acts as a control on animal density.

**Keywords:** SECR; capture-recapture; camera trapping; population ecology; landscape ecology; wildlife management; population modeling; wildlife conservation; elk

## 1. Introduction

In recent decades, camera traps have become increasingly used by researchers and managers to monitor wildlife species and populations [1]. Camera traps are relatively easy to deploy and service over time and often are the most efficient way to collect wildlife presence data with minimal human interference [2]. For species whose individuals are uniquely identifiable through photographs, camera trapping data have also been extended to population estimation through capture-recapture statistical methods [3]. More recently, camera trapping data have been used to estimate the populations of species for which individuals are not uniquely identifiable through photographs with statistical methods that model spatial counts or the presence/absence of species [4].

A relatively recent and potentially powerful addition to the suite of unmarked spatial presence/count models is the unmarked spatial capture recapture (uSCR) model, which estimates population density through individual captures and recaptures across sites where individual identity is an unknown latent variable [5]. A key advantage of uSCR models over other unmarked abundance models is that population density can be directly estimated through the modeling of individual activity centers across a defined state space, as opposed

to needing an independent measure of animal space to define the effective sampling area around the sites [4]. Additionally, explicitly modeling activity centers removes the bias accrued by missing spatial heterogeneity in capture probability due to individual activity centers that vary in distance from count sites [6].

After their initial introduction, Evans and Rittenhouse [7] extended uSCR models to fit spatial, categorical covariates to population density and found that these models resulted in similar inferences on spatial population density compared to marked spatial capture-recapture (mSCR) models. Additionally, Ramsey and colleagues [8] formulated a binomial version of uSCR to model population density from presence/absence data and found that accurate densities could still be estimated under the right conditions. These models and their extensions have advanced our ability to estimate animal densities, but they so far lack the ability to model those densities as a function of continuous environmental variables.

To fill this gap and further advance the applicability and utility of uSCR models, we aimed to complete three objectives in this study: (1). formulate a version of uSCR to estimate population density as a function of continuous environmental covariates across the landscape, (2). employ a Bayesian model selection procedure to evaluate relative support for including different covariates on population density, and (3). compare the results of modeling different definitions of spatial counts and spatial presence/absence that are often derived from camera trapping data. We applied our methods to a Roosevelt elk (*Cervus canadensis roosevelti*) population within the Karuk Ancestral Territory in Northern California as a case study, with a final aim of our study to develop an effective population monitoring program for elk in the region. We expect our extensions to uSCR modeling to enhance the utility of unmarked spatial count/presence models across a wide range of study systems in which individuals cannot be reliably identified and environmental controls on density are apparent.

## 2. Materials and Methods

*Study System*

We conducted our study along the Klamath River within the Karuk Tribe's ancestral territory in Northern California in what constitutes the winter range for elk populations in the area (Lat. 41.3009°, Long. 123.5418° Figure 1, [9]). The rugged mountains and river valleys range from about 123 m to over 2500 m in elevation, but the elk winter habitat falls below 780 m in elevation [10]. There is a floristically diverse community dominated by a diverse coniferous overstory consisting of Douglas-fir (*Pseudotsuga menziesii var. menziesii*), incense cedar (*Calocedrus decurrens*), sugar pine (*Pinus lambertiana*), Jeffrey pine (*Pinus jeffreyi*), and white fir (*Abies concolor*) [11]. The area features a Mediterranean climate of warm, dry, sunny summers and cool, wet, overcast winters [12]. Temperatures range from lows of around 0 °C in December/January to highs of over 30 °C in August. Elk are a culturally important species that traditionally made up a staple food source during certain times of the year and were actively managed by the Karuk [13]. Due to fire suppression, habitat loss, and hunting for meat and hides after Euro-American colonization, nearly all elk were extirpated from the Karuk Tribe's aboriginal territory as early as the 1870s [14]. Beginning in 1985, six Roosevelt Elk from Redwood National Park were re-introduced into Elk Creek in Klamath National Forest. By 1996, 232 Roosevelt elk had been re-introduced into Klamath National Forest and the Marble Mountain wilderness by the US Forest Service and California Department of Fish and Wildlife [10]. At least four elk herds now reside in the Marble Mountains with a population size of potentially up to 3000 [10]. Population estimates in the area are very uncertain, however, due to the difficulty in counting individuals from road or plane surveys in the rugged terrain and dense forest cover.

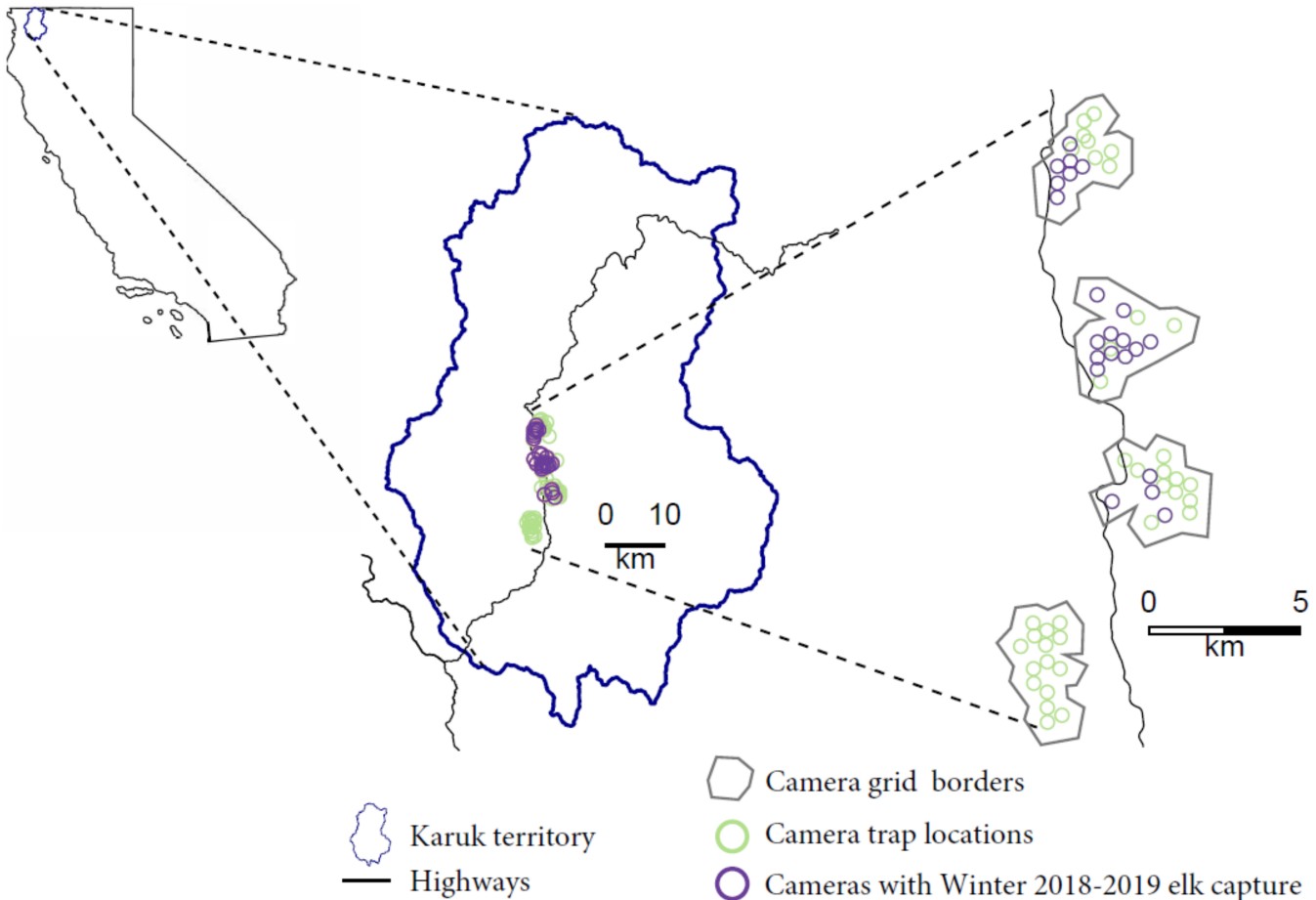

**Figure 1.** Map depicting the study system including Roosevelt elk winter range in a portion of the Karuk Indigenous Territory in Northern California.

1. Camera trapping

We established four hexagonal grid camera arrays in which 15 cameras were placed at the centers of the hexagons approximately 500 m apart (Figure 1). These four grids corresponded to focal areas of the Western Klamath Restoration Partnership's Somes Bar Integrated Project Area and were designed to monitor wildlife responses to forest management actions [15]. Cameras were placed at each site to maximize the likelihood of capturing wildlife by placing the camera at approximately shoulder-height and directing the viewshed along game trails and/or areas with good visibility. We used Bushnell TrophyCam HD Aggressor cameras set to take two photographs at the detection of any movement under the low sensitivity setting with a medium shutter speed and a 'sleep time' of 3 s between photograph bursts. To capture elk activity within a single winter season, we ran the cameras from the end of November 2018 to the end of March 2019.

2. Elk spatial presence/counts

We grouped all photographs of elk into a single record table using the R package camtrapR [16]. For binary presence absence datasets, we created detection histories summarizing elk presence or absence at each camera for each day of the study. To derive count datasets, we investigated two methods of defining the daily detection histories. First, we counted every 'independent' detection of elk in a 24-h period, defined by subsequent images of one or more elk separated by at least 15 min. Second, we took the highest number of individuals seen in a given photograph during each hour of the day and summed those counts to arrive at a 'total count' response. These methods each come with their own labor requirements and levels of risk of double counting or undercounting individuals. Specifically, the 'independent detection' dataset was relatively easy to derive using camtrapR,

while the 'total count' dataset involved more image sorting labor and the writing of custom R functions. In the 'independent dataset,' the overcounting of individuals can occur if a single animal remains in front of a camera for more than 15 min or returns to the camera more than 15 min later in the day, while undercounting individuals can occur if there are more than one individual in a given photo or multiple individuals get captured by the camera in separate photos within 15 min of each other. We chose 15 min as opposed to a longer interval for photo independence because our camera traps were generally placed on game trails facilitating animal movement through the area and not feeding/resting sites. In the 'total count' dataset, overcounting individuals can occur if the same group of individuals remain at or are captured at a given camera during separate 1-h intervals in a day, and undercounting individuals can occur if different groups of individuals are captured at a given camera during the same 1-h interval in a day. The risk of undercounting individuals is much higher in the 'independent detection' dataset, while the risk of over-counting individuals is not substantially higher in the 'total count' dataset because counts are based on known different individuals in single photos. It is also important to note that uSCR models were formulated with the assumption that double counting of individuals in the observation dataset may occur [5]. A priori, we thus expected the 'total count' dataset to be most appropriate for uSCR modeling but wanted to evaluate performance of the easier to derive 'independent count' and 'presence-absence' datasets.

3. GPS collaring

We used data collected from four elk (three females and one male) in our study area to derive an independent estimate of elk space use to serve as a prior in our Bayesian models and a comparison for our model results (see next section). Elk were captured in the winter of 2020 using baited clover traps and collared without sedation. Animals were captured according to a collaring plan approved by the California Department of Fish and Wildlife and the Karuk Tribe. Each elk was fitted with Vectronic Vertex Lite 2D GPS collars set to transmit location fixes every two hours (Vectronic Aerospace GmbH, Berlin, Germany). To estimate space use, we used kernel density estimation (KDE) using a reference bandwidth and bivariate normal kernel to derive a 95% home range polygon for the winter season using the 'adehabitat' R package [17]. To convert from home range estimates to the capture-recapture sigma (space-use, see next section) parameter, we divided the radius (assuming a circular home ranges) of our home ranges by 2.45 [18].

4. Unmarked spatial capture recapture model

To estimate population size from spatial counts, we followed the methods set out in Royle and Chandler [5]. In their model, spatial counts at capture sites are used as the observed Poisson-distributed response of an unobserved population of individuals with specific activity center locations. The members of this population are drawn from an artificially large superpopulation through a data augmentation approach with their inclusion in the latent population and their activity centers explicitly tracked through Montel Carlo Markov Chain (MCMC) iterations allowing for Bayesian inference [19]. We set this superpopulation size at an unrealistically large population size of $N = 2000$ to avoid biasing the model estimates low. The standard mSCR formulae for modeling density as a function of detection probability (g0) and distance to activity centers (sigma) then apply [5,20]. Specifically, g0 is the probability of detecting an individual at its home range center, and sigma is a scaling parameter that describes how this probability of detection declines in a half-normal distribution as the distance from the individual's activity center increases (Figure S4, [20]). We also fit Ramsey et al.'s [8] presence-absence uSCR formulation, with spatial presence-absence modeled as an observed binomial-distributed response of the unobserved population.

We extended these models to incorporate continuous covariates across the landscape as predictors of activity center density. We did this by explicitly including a 500-m state space grid on which individual activity centers could vary between MCMC iterations. We buffered the cameras by 3000 m to capture the vast majority of potential activity centers of

elk captured at the camera traps. As spatial covariates, we used Landsat-derived vegetation data and elevation (Rangeland analysis platform, RAP [21]). Specifically, we used total percent forbs and grasses cover by summing the RAP estimates of the annual forbs and grasses and the perennial forbs and grasses per cell, and percent tree cover. We then aggregated these 30 m-resolution data to 500 m using the mean value of included cells to match our state-space grid and modeled individual activity center locations on the 500-m grid cells as a response to forbs and grasses cover and elevation. The Pearson's correlation coefficient between all variables was less than 0.6. In addition to the environmental covariates, we included an intercept for activity center density. We wrote the model in the Bayesian Nimble coding language within R [22,23]. We used uninformative flat priors for all model variables except the sigma (space use) variable described above, on which we placed an informative prior based on the GPS-collar data following a normal distribution with mean = 1 km and standard deviation = 0.5 km.

In addition to incorporating continuous covariates, we also included a Bayesian model selection component to evaluate support for including different covariates in the model. Specifically, we included an indicator variable per covariate that controlled whether that covariate was included and updated in a given iteration formatted as a reversible jump MCMC (RJMCMC, [24]). We ran this model using the three different definitions of spatial count/presence-absence and compared the results, each for 80,000 MCMC iterations after a burn-in of 20,000 iterations.

## 3. Results

All 60 game cameras were operational across the entire winter of the 2018–2019 study period. A total of 22 of the 60 cameras detected elk on at least one occasion. Summarized as counts of independent detections per occasion, the response variable at those cameras varied from 1 to 26 elk across occasions. Summarized as total counts of all individuals detected in photographs per occasion, the response variable varied from 1 to 48 elk across occasions. Finally, the presence-absence response variable varied from 1 to 11 elk across occasions. Our computed 95% KDE winter home range polygons of our four collared elk had a mean of 18.96 km$^2$ and a standard deviation of 6.67 km$^2$.

The Bayesian uSCR models based on count responses and modelled with a Poisson distribution converged relatively well and had similar results (Table 1, Figures 2, S1 and S2). The model based on a presence-absence response fit to a binomial distribution did not converge well and differed substantially in posterior inference from the count models (Table 1, Figures 2 and S3). The probability of detecting an individual at its activity center was estimated to be 0.1 in the 'independent detection' Poisson model, 0.34 in the 'total count' Poisson model, and 0.05 in the 'presence-absence' binomial model. The spatial scaling parameter sigma was estimated to be 0.26 km in the 'independent detection' Poisson model, 0.23 km in the 'total count' Poisson model, and 1.09 km in the 'presence-absence' binomial model. The count models provided an estimated mean density of around 0.5 elk/km$^2$, a positive effect of forbs and grasses cover on density, a negative effect of tree cover on density, and a marginal and variable effect of elevation on density (Table 1). The binomial presence-absence model provided an estimated mean density of 0.1 elk/km$^2$ and negative effects of all three environmental covariates on elk density. Only the uSCR model using the total count of individuals as a response resulted in posterior distributions with significant credible effects (95% CI not overlapping 0) of environmental covariates on density, and not for the elevation covariate. In all the models, there were high posterior inclusion probabilities for the vegetation covariates and a low inclusion probability for the elevation covariate. Predictions made from the posterior environmental covariate posterior mean coefficients from the uSCR model using the total count of individuals as a response indicated substantial spatial variation in elk density across the landscape (Figure 3).

**Table 1.** Posterior parameter estimates for each uSCR model.

| Parameter | Mean (Probability of Inclusion in Parentheses, If Applicable *) | | | Standard Deviation | | | 95% Lower CI | | | 95% Upper CI | | |
|---|---|---|---|---|---|---|---|---|---|---|---|---|
| | Independent Detection Count | Total Individual Count | Presence-Absence | Independent Detection Count | Total Individual Count | Presence-Absence | Independent Detection Count | Total Individual Count | Presence-Absence | Independent Detection Count | Total Individual Count | Presence-Absence |
| g0 | 0.100 | 0.340 | 0.050 | 0.023 | 0.099 | 0.028 | 0.062 | 0.142 | 0.023 | 0.149 | 0.534 | 0.125 |
| sigma | 0.264 | 0.225 | 1.09 | 0.024 | 0.017 | 0.682 | 0.219 | 0.201 | 0.227 | 0.313 | 0.269 | 3.115 |
| Density | 0.463 | 0.566 | 0.109 | 0.167 | 0.157 | 0.191 | 0.206 | 0.301 | 0.003 | 0.858 | 0.907 | 0.838 |
| FG cover | 0.052 (0.73) | 0.076 (0.908) | −0.035 (0.529) | 0.045 | 0.027 | 0.047 | −0.026 | 0.015 | −0.121 | 0.130 | 0.125 | 0.066 |
| Percent tree cover | −0.081 (0.81) | −0.116 (0.991) | −0.065 (0.712) | 0.066 | 0.046 | 0.053 | −0.199 | −0.194 | −0.155 | 0.043 | −0.017 | 0.070 |
| Elevation | −0.004 (0.17) | $1.2 \times 10^{-4}$ (0.003) | $-4.9 \times 10^{-4}$ (0.010) | 0.003 | $3.0 \times 10^{-4}$ | $7.2 \times 10^{-4}$ | −0.011 | −0.001 | −0.002 | 0.001 | $3.8 \times 10^{-4}$ | 0.001 |

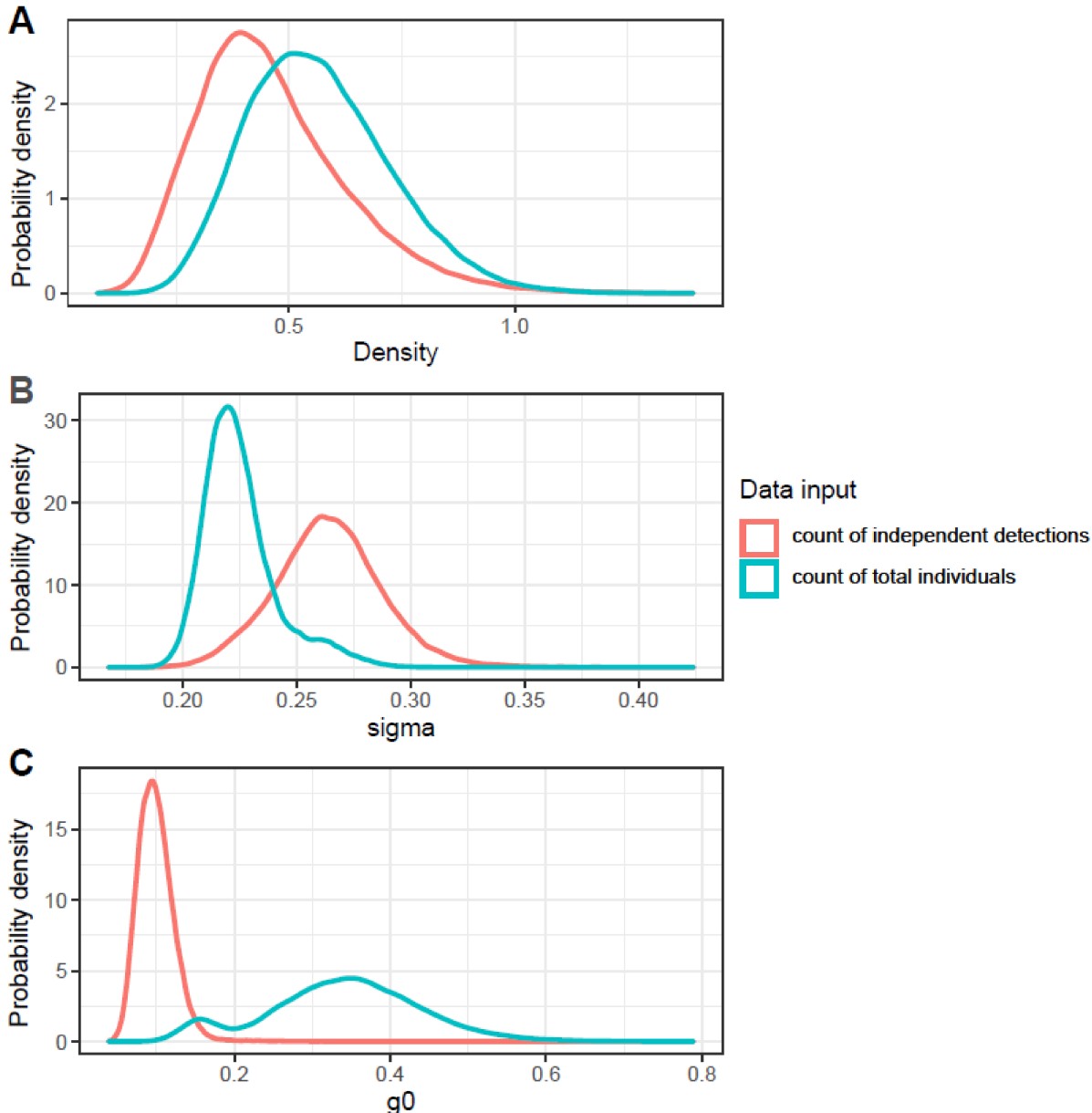

**Figure 2.** Posterior probability density distributions of (**A**) Roosevelt elk density, (**B**) space use (sigma, in km), and (**C**) probability of detection (g0) parameters from the uSCR model MCMC chains. The presence-absence binomial version of the model is not presented here due to poor chain convergence (see Supplementary Information).

# Predicted elk density

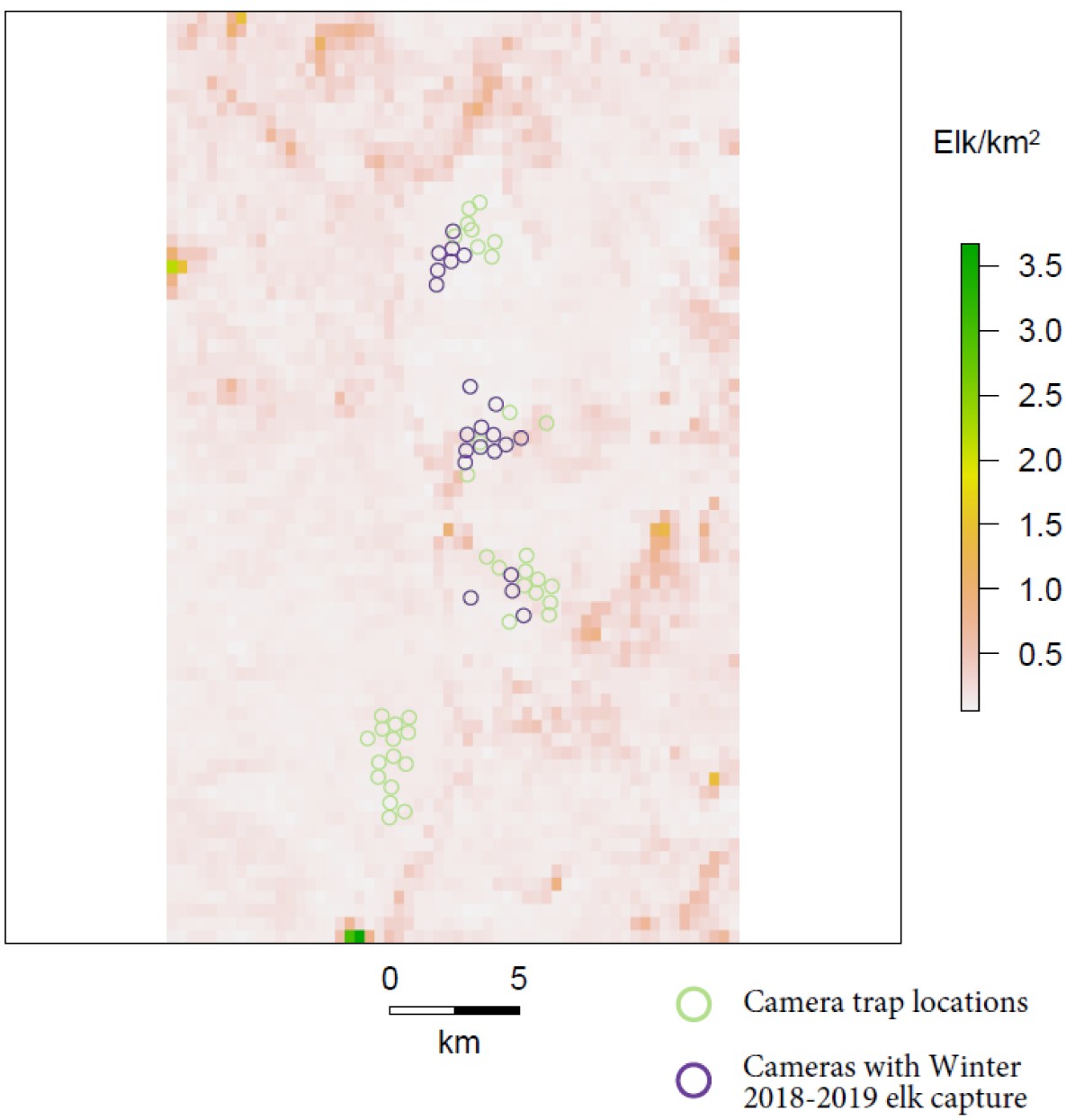

**Figure 3.** Predicted Roosevelt elk density across the study area based on posterior estimates of the percent forbs and grasses and tree cover covariate coefficients derived from the uSCR model fit to total counts of elk at each camera station.

## 4. Discussion

Our results suggest that the extensions to uSCR modeling that we implemented were successful—by modeling activity centers across a grid with associated environmental covariate values per cell, we directly drew inference on the effects of those covariates on density. Doing so comes with several advantages over previous iterations of uSCR models. Specifically, Chandler and Royal's [5] original formulation allowed for tracking of animal activity centers and a posterior estimate of density across space, but they did not directly model the effects of environmental covariates on density. Evans and Rittenhouse [7] extended the original uSCR model to include categorical environmental covariates on density, but did not include support for continuous covariates. There are distinct advantages to

using continuous covariates in analyzing ecological and landscape-scale phenomena as they contain much more information than broadly binned categories [25,26]. This is an especially important consideration when the modelled response variable, the animal density in this case, is itself continuous [27]. Thus, our extensions to allow uSCR models to estimate the effects of continuous covariates on animal density represent an important progression with wide applicability. It would also be straightforward to fit categorical covariates using our formulation.

The inferences drawn from our uSCR models on environmental effects on elk density mostly followed expectations and reflect existing knowledge on elk ecology. Specifically, there was a positive credible effect of forbs and grasses cover on density and a negative credible effect of tree cover on density. Though generalists, there is evidence that elk are preferential grazers [28] and, thus, a positive density response to increased grass and forb forage availability makes sense. The negative response of elk density we found to tree cover in the forested landscape of elk wintering grounds along the Klamath bucks classic views in ungulate management was that canopy-induced thermal cover is an important factor for cervid condition and survival over winter seasons [29,30], but our results follow experimental research that winter energetic costs for elk in fact increase with increasing tree cover [31]. Elk's potential need for thermal cover is likely even less in the relatively mild winter conditions along the Klamath River in Karuk Indigenous territory. Although we do not have independent robust estimates of elk population density in the area sampled by our camera traps, general knowledge of elk populations in the area suggest that our uSCR estimates are reasonable [10]. While the direction of the estimated environmental covariate effects on elk density and mean density estimates make ecological sense and we think that our predicted density map likely reflects true patterns in elk distribution on the landscape, that density map also indicates that there may be some issues with our model's spatial predictions at extreme values of the environmental covariates (very high grass/forb cover and/or very low tree cover). Specifically, two cells at the southern edge of our study area had very high predicted densities of more than 3 elk/km$^2$. It would be interesting to visit those sites to determine if there is evidence of heavy elk use and to ground-truth the environmental conditions there.

Although our environmental covariate coefficient and elk population density estimates followed expectations, the sigma (space use) parameter did not. Even with an informed prior based on GPS telemetry data gathered from several individuals in the area, the uSCR model estimates of sigma trended downward in a pattern similar to that found by Ramsey et al. [8]. Due to this bias, we argue that these models should not be used to make inference on animal space use. The fact that population density estimates were reasonable suggests that the uSCR models did a good job in balancing the probability of detection and space use parameters to derive population density estimates, but that the detection and space use parameters themselves were likely not realistic. In our case, the probability of detecting an elk was likely lower and elk space use likely higher than estimated in the uSCR models.

There were also important differences between our uSCR models based on different definitions of the response variable and statistical distributions used to model that response. These differences were minimal between the two definitions of spatial counts, suggesting that summing 'independent' detections across an occasion approximated the desired count response well. This method has been used in the literature before [7] to good effect, and our results increase the body of evidence that summing 'independent' detections per occasion can work reasonably well. That said, our uSCR model based on a stricter count response that considered multiple individuals in photographs had more significant covariate effects on density and better convergence of the MCMC chains. Furthermore, counting all individuals in photographs over an occasion, even at the risk of double counting, better represents the data requirements of the original uSCR formulation proposed by Chandler and Royle [5]. The presence/absence version of uSCR had comparatively poor results, with posterior population estimates severely right skewed, drastically low, and with worse chain convergence. We posit this is likely due to the exacerbated loss of information in

reducing counts to presence/absence in a herding species. Given the results of our multiple tests of response variable definitions and statistical distributions, we recommend that strict spatial counts that consider multiple animals in a given detection should be used with a Poisson-formulated uSCR model where possible, but that sums of 'independent' detections within each occasion may suffice in certain situations.

In addition to our extensions to uSCR modeling and testing of multiple definitions of the response variable, our successful use of RJMCMC as a form of variable selection is worth discussing. Although not commonly considered as a potential method of model selection in SCR models [32], RJMCMC comes with some distinct advantages. First, it allows a variable to be turned off from both its inclusion in the model and proposed step updates, preventing it from wandering widely in parameter space while not included in the model [24]. Second, a variable's posterior inclusion probability is a direct measure of the estimated support for including it in the model and is available after a single MCMC run. Due to the often-intensive computational requirements of fitting Bayesian models, a model selection procedure that does not require multiple MCMC runs with different variable combinations comes with huge computational and time advantages. We argue that RJMCMC should be more widely considered as a variable selection procedure in Bayesian capture recapture models, as well as Bayesian ecological models more generally.

Additional extensions to the uSCR modeling framework could increase its utility even further. For species for which sex is identifiable through whatever method of detection is used, sex-specific detection and space use parameters could be estimated [33]. This specificity may help the uSCR models converge on more accurate detection and space use estimates, and sex-specific inferences are important for many ecological research and management efforts [34]. While we could have reliably identified sex on elk adults from photographs throughout our study area for most of the winter, there is a short period between the time that bulls drop their antlers and the time their regrowth can be reliably seen when sex identification is difficult. Additionally, it is impossible to reliably identify the sex of elk calves from photographs, leading us to group sexes in our models. It would also be possible to add other covariates to detection probability to better account for uneven detection probabilities across space/detection stations [35]. Finally, incorporating uSCR-based estimates of different age classes could fit very well into integrated population modeling efforts that aim to separately estimate the density of juvenile and adult animals [36].

## 5. Conclusions

Our modifications of uSCR models allow for their estimation of continuous environmental covariate effects on wildlife population density. As all wild populations live in heterogeneous environments, our methods can be applied widely to populations for which spatial counts can be derived to better understand their ecology and more effectively manage them. Additionally, understanding environmental effects on population density may allow for the effective extrapolation of the uSCR models to unsampled areas where similar environmental relationships with population density occur. Although uSCR models comes with distinct advantages over other unmarked abundance modeling frameworks and our extensions increase their utility, it is important to consider what modeling framework is most appropriate for a given detector array and species [4]. For example, uSCR models rely on spatial autocorrelation in detections so that one individual may be detected at multiple sites [8]. More research is needed to determine the relative tradeoffs between violating assumptions of independence between sites in some unmarked abundance modeling frameworks (N-mixture, Royals-Nichols) vs. inadequate spatial autocorrelation between sites in uSCR models [8,36,37]. Further comparisons of methods in simulated and/or well-studied populations are needed to parse out differences between methods and the situations in which each are most appropriate [4]. We hope that our addition of continuous spatial covariates to the uSCR modeling framework will increase the utility of uSCR models in a wide array of research and management efforts.

**Supplementary Materials:** The following are available online at https://www.mdpi.com/article/10.3390/rs14051087/s1, Figure S1: Trace plots and Monte Carlo standard error as a percentage of the posterior SD (MCEpc) values for variables in models of counts of elk independent detections. The trace plots and MCEpc values indicate adequate mixing of most variables but problematic mixing of the environmental covariate variables (chains not clearly converged and MCEpc values over 5). Figure S2: Trace plots for variables in models of total counts of elk individuals. These trace plots and MCEpc values indicate adequate mixing of all variables, though the MCEpc values for the A.FG and A.Tree environmental covariate coefficients are high. Figure S3: Trace plots for variables in models of presence-absence of elk. Trace plots for variables in models of total counts of elk individuals. These trace plots and MCEpc values indicate inadequate mixing of all variables, except g0. Figure S4: The effects of increasing distance from an individual's activity center on the probability of detecting that individual, parameterized by sigma. This plot represents the relationship estimated by the model fit to the 'total count' elk dataset.

**Author Contributions:** Conceptualization, T.C., W.D., E.T., W.T.B., L.M., and J.B.; methodology, T.C., and W.T.B.; software, T.C., and W.T.B; validation, T.C., and J.B.; formal analysis, T.C.; investigation, T.C.; resources, W.D., W.T., W.T.B, L.M., and J.B; data curation, T.C., E.T., B.J.S., J.C., and A.D.; writing—original draft preparation, T.C.; writing—review and editing, all authors.; visualization, T.C.; supervision, W.T., L.M., and E.T.; project administration, E.T., W.T., J.C., D.S.-W., L.M., and J.B.; funding acquisition, E.T. and D.S.-W. All authors have read and agreed to the published version of the manuscript.

**Funding:** This research was funded by the California Big Game Management Program Grant Agreement Number P1780108.

**Data Availability Statement:** The code needed to run our models will be published in the Supplementary Information. The camera locations and elk presence data are sensitive and available on request and the approval of the Karuk Tribe.

**Acknowledgments:** We would like to thank David S.L. Ramsey for his assistance in developing the Nimble code needed to run the models.

**Conflicts of Interest:** The authors declare no conflict of interest.

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
