# Peer review of "Estimating Wildlife Density as a Function of Environmental Heterogeneity Using Unmarked Data"

_remotesensing, doi:10.3390/rs14051087_

Round 1

Reviewer 1 Report

The manuscript deals with interesting possibility to uncover real density of targeted species (Roosevelt elk Cervus canadensis roosevelti) based on camera recordings using unmarked individuals. The authors further used structure of the key vegetation layers based on remote sensing to estimate the distribution of the species within the wider area of Northern California. To enhance their estimation, the authors included into their model also home range size based on data from their study area. I feel that all these indirect methods may suffer from some shortcuts, but in the case of the elks I suggest that this methodology may bring quite reliable results. The manuscript is well prepared, and a lot of details are put about the methodology used. However, I think that for the broader readers it can be useful to explain in more details meaning of variables that are shown in Appendix SI. I have also some minor comments to the manuscript:

Introduction – I lack the scientific name of Your study species here

P2, L74 – Please, remove redundant space before “By”

P4, L164 – I would use the word “elks” instead of “elk” for plurals. Please check also other parts of the manuscript

P5, L179 - Please, remove redundant space before “Predictions”

P4, L269-278 – The authors stated that the method can be extended when the sex of the species can be determined. Since Your cameras were installed during the winter, I suggest that at least some males were possible to determine according to presence of antlers. My question is whether this statement was applicable also for Your study. At least, I am missing the statement that this was not applicable for Your study, because part of males during the study period might already lost their antlers.

P9, L360 – Remove redundancy space

Reviewer 2 Report

In their manuscript entitled "Estimating wildlife density as a function of environmental heterogeneity using unmarked data" Connor *et al.* present an extension of the uSCR modeling framework in order to estimate animal density as a function of continuous environmental variables. They also use an elk population along the Klamath River in Northern California (within the Karuk Indigenous Territory) as a case study.
Their declared aim is: i) To develop a new version of the uSCR model for the elk monitoring program in the region and ii) To provide an evaluation of its results by comparing the posterior parameter estimates.

In my opinion, both goals are reached and the authors provided a detailed description of  methodologies, statistical analyses, and final results. In their introduction, the authors describe the current state of the research and define the main aim of the work. The Discussion section develops a straightforward discussion of the authors' findings, although some interpretations of the results lack a solid foundation (see below). However, the manuscript could be of interest to the audience of the Remote Sensing journal, and I suggest its acceptance after a minor revision.

## Minor and specific point-by-point comments follow:

- Please check throughout the text that before using an acronym you put the explanation *in extenso* (e.g. L128. Explicit that 'MCMC' stands for 'Markov chain Monte Carlo')
- L80-83. I suggest removing the sentence from Material and Methods and moving it in the Introduction section
- L89. In Fig.1 I can count more than 15x4 camera trap locations. But in the Results section, the authors reported the activity of 60 game cameras. Please clarify.
- L92. The two references [14,15] seem to point to the same report. Please check.
- L107-108. The authors stated that their count method has a potential bias coming from the risk of double-counting individuals. What is the magnitude of this potential bias and what do the modellers can do to mitigate its effects? Please discuss.
- L160. In Fig. 1 I can count at least 25 camera locations with elk captures, but the authors reported only 21. Please clarify.
- L168-169. I guess that supplementary materials are reported with incorrect labels. e.g. Appendix A instead of Appendix S1, and Figg A1,A2,A3 instead of S1,..., please check. Moreover, I would consider including in the caption a brief description of the importance of trace plots and MCEpc values to make them understandable to scientists working outside the topic of the paper.
- L173. The authors state that the count models showed a negative effect of elevation on density, but this is true for the independent detection count version only. Please clarify.
- L175-177. Why are the authors so confident in stating that 'Only the uSCR model using the total count of individuals as a response resulted in posterior distributions with significant credible effects of the environmental covariates'?
- L179. The results section lacks any presentation of the results obtained for the parameters space use (sigma) and probability of detection (g0). Please insert a new sentence, highlighting the differences obtained from the two versions of the uSCR model.
- L179-182 and Fig.3. Predicted high values of elk density occur in only 3 marginal cells on the map, away from the camera trap locations. Did the authors explore on the field these cells to confirm their predictions? In lines 231-236 the authors highlighted their doubts about the estimates of the probability of detection and space use parameters themselves, although they believe the model has a good balance. By contrast, at the camera trap locations with elk capture, density values appear to be far below 0.5. Are the authors confident that the spatial variation depicted in the map is robust enough to represent the actual distribution of elk density in the Karuk Territory? Why did they report the total count version only? Please clarify.
- Figure 1 - Please, number and outline the four hexagonal grid camera arrays. The boundary line of Karuk territory is of two different colors in the map and in the legend, please verify.
- Figure 2 - Please, insert a tag name for each plot (es. A, B, C) pointing to the caption; replace 'D' with 'Density' and 'Probabiilty' with 'Probability'.

---

Reviewer 3 Report

Reviewer Comments

The manuscript is titled: “Estimating wildlife density as a function of environmental het-2 erogeneity using unmarked data”. It is a Research article which focuses on a species’ distribution parameter (the density). The manuscript is publishable after major revision. There are some imperfections which must be improved before the acceptance of this manuscript, such as:

  1. The manuscript lacks clear hypotheses/objectives.
  2. The introduction is average
  3. The mathods lacks the study area description
  4. The results are well synthesized and bring out some relevant inferences about the study’s methods

      3. The discussion is averagely well presented

Some Issues

  1. Line 76: Write Department instead of Dept.
  2. The study area is not described: many ecological parameters are not brought out: temperature, precipitations, sunshine, vegetation cover, geographic coordinate, seasons and their duration, elevation, etc. All these parameters have influence on species’ ecology.
  3. Line 105: the time for independent photos is 15 minutes only, I presume that it is too stringent, since most studies I have read and those I undertaken always use 30 minutes or one hour.
  4. Line 137: Not all your readers know MCMC, it is better to write it in whole the first time then authors can abbreviate it after.
  5. Why authors used only continuous covariates?
  6. Line329 and 374: Why the references 5 and 29 are written in up case letters?

Reviewer 4 Report

Whilst by its nature rather technical, this is still an interesting and pertinent paper. However, you could perhaps ease the readability for the non-technical researcher interested in mammal and esp deer populations and assessments etc. So perhaps some plain English expanations of technical jargon and the relevance of what you found would be a big improvement. 

A minor point is to please use 'photograph(s)' not 'photo(s)' - it just looks a little lazy! 

Also, data is plural - i.e. data 'are' not data 'is'. 

A short plain English 'Conclusions with recommendations' would also enhance the paper's usefulness.
